# *Calendula officinalis* Triterpenoid Saponins Impact the Immune Recognition of Proteins in Parasitic Nematodes

**DOI:** 10.3390/pathogens10030296

**Published:** 2021-03-04

**Authors:** Maria Doligalska, Kinga Jóźwicka, Ludmiła Szewczak, Julita Nowakowska, Klaudia Brodaczewska, Katarzyna Goździk, Cezary Pączkowski, Anna Szakiel

**Affiliations:** 1Department of Parasitology, Faculty of Biology, University of Warsaw, 02-096 Warsaw, Poland; kinga.jozwicka@gmail.com (K.J.); ludmila.szewczak@wihe.pl (L.S.); kbrodaczewska@wim.mil.pl (K.B.); kgozdzik@biol.uw.edu.pl (K.G.); 2Laboratory of Electron and Confocal Microscopy, Faculty of Biology, University of Warsaw, 02-096 Warsaw, Poland; julita@biol.uw.edu.pl; 3Department of Plant Biochemistry, Faculty of Biology, University of Warsaw, 02-096 Warsaw, Poland; myhacp@biol.uw.edu.pl (C.P.); szakal@biol.uw.edu.pl (A.S.)

**Keywords:** triterpenoid saponins, TEM nematode ultrastructure, protein patterns

## Abstract

The influence of triterpenoid saponins on subcellular morphological changes in the cells of parasitic nematodes remains poorly understood. Our study examines the effect of oleanolic acid glucuronides from marigold (*Calendula officinalis*) on the possible modification of immunogenic proteins from infective *Heligmosomoides polygyrus bakeri* larvae (L3). Our findings indicate that the triterpenoid saponins alter the subcellular morphology of the larvae and prevent recognition of nematode-specific proteins by rabbit immune-IgG. TEM ultrastructure and HPLC analysis showed that microtubule and cytoskeleton fibres were fragmented by saponin treatment. MASCOT bioinformatic analysis revealed that in larvae exposed to saponins, the immune epitopes of their proteins altered. Several mitochondrial and cytoskeleton proteins involved in signalling and cellular processes were downregulated or degraded. As possible candidates, the following set of recognised proteins may play a key role in the immunogenicity of larvae: beta-tubulin isotype, alpha-tubulin, myosin, paramyosin isoform-1, actin, disorganized muscle protein-1, ATP-synthase, beta subunit, carboxyl transferase domain protein, glutamate dehydrogenase, enolase (phosphopyruvate hydratase), fructose-bisphosphate aldolase 2, tropomyosin, arginine kinase or putative chaperone protein DnaK, and galactoside-binding lectin. Data are available via ProteomeXchange with identifier PXD024205.

## 1. Introduction

Parasites are often susceptible to natural compounds, suggesting that plant products may enhance, or eventually replace, chemical drugs in the antiparasitic treatment of human and animals [1,2]. Although it is now possible to artificially synthesise many naturally occurring active factors, plants remain a ready source of bioactive products such as tannins, flavonoids or saponins [3,4]. In spite of growing in vivo and in vitro data indicating that a number of plant products, crude extracts and their derivatives have anthelmintic activity, only a few plant species have been confirmed to demonstrate specific activity against different nematode species [5,6]. By binding to the cuticle proteins and by changing their physical and chemical properties, condensed tannins may delay the exsheathment of the nematode infective larvae [2,7]. The distinct activity of these compounds against parasites seems to result from specific targeting at the molecular level.

One group of plant-derived factors that has also aroused considerable interest from the veterinary and biomedical sciences due to their positive and negative effects on various organisms are the pentacyclic triterpenoids; however, their effect on cell structure remains largely uncharacterised [8]. A fuller understanding of these plant products would support the design of biodegradable and effective drugs for livestock and food production.

Due to their detergent properties and ability to increase cell permeability, as well as their cytotoxic and cytostatic activities, saponins offer great potential as therapeutic agents [9]. Marigold (*Calendula officinalis* L, family Asteraceae), a widely used plant in traditional veterinary and medicine, contains two series of oleanolic acid (3β-hydroxy-olea-12-en-28-oic acid, OA) glycosides, these being *glucuronides*, derivatives of 3-O-monoglucuronide F, and *glucosides*, derivatives of 3-O-monoglucoside I. The glucuronides comprise pentacyclic triterpenoid saponins and oleanolic acid glycosides [10].

The *Heligmosomoides polygyrus bakeri* infection model in laboratory mice is representative of the gastrointestinal nematode parasitic infection in ruminants [11]. The nematode life cycle comprises two phases: in the first, nematode eggs hatch in faecal material and pass through two moults to reach the infective L3 stage; in the second phase, taking place in the host, the larvae develop into L4 which after moulting move into the mucosa and mature. The adult females release eggs, which leave the host in expelled faecal material and later germinate.

Physiological processes are strictly regulated at the transcriptome level, and a wide range of factors impacting hormonal or enzymatic activity influence larval growth and development via the regulation of anabolic pathways. Larvae require several enzymes for collagen synthesis and removal of the existing cuticle to form a new cuticle [12]. Heat shock proteins are also synthesised in response to stress conditions to regulate degradation or refold denatured forms [13]. Another key function in these processes is played by mitochondria, whose activity depends on the stage of development of the nematode and oxygen availability [14]. Of these, the external part of the nematode life cycle is still regarded as the most effective point at which to control the parasites in pasture [15].

P-glycoproteins (Pgp) are ATP hydrolysis-dependent transporters, members of the ABC transporter superfamily. They are known to reverse anthelmintic activity, participate in cell detoxification and protect the parasite from xenobiotics and host defence molecules [16,17]. By destroying cell membranes and affecting lipid content or by denaturing the metabolic and structural proteins associated with Pgp function, natural products can serve as effective inhibitors of nematode infectivity [18].

The infective heligmosomoides larvae produce proteolytic enzymes when penetrating the host mucosa; these play a key role in successful infection by assisting the settlement of larvae in the tissue. The nematode epicuticle, rich in lipids and associated with a glycoprotein-rich surface coat, sustains an immune active surface and regulates the recognition of the innate immune system cells [19,20,21,22]. The use of saponins may attenuate the virulence of the nematode by changing its immunogenic protein profile [20], thus impairing successful invasion.

Although proteomic analyses of *H. polygyrus bakeri* have partially clarified the factors inducing the immunoregulative mechanisms of the adult or L4 stages [22,23,24,25], the molecular strategy used by larvae to evade the host innate immune system remains unknown. Triterpenoid saponins have been found to reduce the infectivity of the L3 stage by altering its glycoprotein pattern [20]. Although previous studies have demonstrated that triterpenoid saponins in marigold products have non-specific destructive activity [26], proteomic studies may identify specific potential nematicidal action.

The aim of the present study was to determine the effects of glucuronides of oleanolic acid (GlcUAOA) extracted from *Calendula officinalis* flowers on the ultrastructure of infective *H. polygyrus bakeri* L3 larvae and on the profile of immunogenic proteins of their somatic extract.

## 2. Results

### 2.1. Changes in the Ultrastructure of L3 Evaluated in TEM

*Heligmosomoides polygyrus bakeri* larvae cultured from the eggs to L3 stage for 10 days were harvested and used for Transmission Electron Microscopy (TEM) analysis. Transverse and longitudinal sections of the anterior part of the larvae were taken and examined. Changes were observed in the ultrastructure of the cuticle, hypodermis, muscle cells, nuclei and mitochondria (Figure 1).

Sections of the L3 control larvae, L3(CTR), are shown in (A), (B), (C) and (D). The epidermis of the larvae cultured with ethanol L3(EtOH) had shrunk away from the cuticle (E), (F), (G), (H). The distance between the exo- and endocuticle was six times greater than in control larvae, and the cuticle was more lucent and swollen in a few locations (E), (G). Aggregated and condensed heterochromatin could be seen in the nuclei of hypoderm cells (F), (G). While the myofibril arrangement was regular in the control larvae, running parallel to the long axis of the muscle fibre (C), these became irregular, shifted and discontinuously arranged after ethanol treatment (G).

Ethanol caused a separation of the cuticle layers (E), but saponins dissolved in ethanol seem to revert this effect (I). The larvae treated with GlcUAOA demonstrated changes in the position of the cuticle, and the cuticular layers adhered tightly to one another (I), (J). The hypoderm cells contained destroyed nuclei and resembled activated apoptosis (I), (J), (K). The cells displayed several ultrastructural features typical for apoptotic cell death, such as surface cell blebbing and nuclear chromatin marginalisation and aggregation. The nuclei were found to be disintegrated and fragmented into distinct spherical fragments with highly dense chromatin, suggesting the formation of apoptotic bodies in late apoptosis (I), (J), (K). Cells in the late stage of necrosis also displayed nucleoskeletal structures, the nuclear envelope was absent, and the chromatin was superaggregated and condensed (I), (J), (K). The mitochondria were very large and swollen, with an amorphous matrix and prominent lucent vesicles; the mitochondrial cristae were disintegrated and deleted (L). No regular arrangement of myofibrils was observed; in one muscle cell, both transverse and oblique myofibrils were noted (K). Additionally, the nucleus of muscle cells had shrunk and the chromatin was strongly condensed (K). The cytoplasm of muscle cells vacuolated and was filled with prominent myelinated structures which were probably mitochondria that had undergone autophagy (J), (K), (N).

TEM images (scale bar 500 nm: (D), (H), (M)) show details in the structure of nematode cuticle exposed to EtOH and GlcUAOA. In the control larvae, the cuticle was found to tightly cover the body of the parasite. Several layers were shown, viz. the epicuticle, cortical zone, medial zone and basic zone (D). In larvae exposed to EtOH, the cortical zone of both the old and new cuticles demonstrated swelling (H); in the early stage of apolysis, the old cuticle was found to separate from the new one. In larvae exposed to GlcUAOA, the epicuticle, cortical zone and medial zone are thickened and peeled of the basal zone (M).

All ultrastructural changes in the larvae were verified by morphometric analysis (Figure 2).

A significant reduction in the length of myofibrils was observed: from 3296 ± 536.9 nm in L3(CTR) to 1605 ± 708.9 nm in L3(EtOH) and to 949 ± 337.9 nm in L3(GlcUAOA) (Figure 1C,G,K, respectively, and Figure 2A). Ethanol treatment resulted in the cuticle layers separating: the distance between the hypoderm surface and cuticle measured at the cross section of the larvae increased six fold (Figure 2B) from 53.8 ± 52 nm in L3(CTR) (Figure 1D) to 318.2 ± 69 nm in L3(EtOH), (Figure 1G,H) and to 135.9 ± 32 nm in L3(GlcUAOA), (Figure 1M).

The mitochondrion was twice as large in L3(GlcUAOA) as in L3(CTR) (Figure 1B,J,L): its length increased from 727.5 ± 410 nm in L3(CTR) to 1251.0 ± 178.0 nm in L3(GlcUAOA), and its width grew from 327.0 ± 76.2 nm in L3(CTR) to 870.5 ± 187.6 nm of L3(GlcUAOA). The area of the mitochondrion, calculated by multiplying its length by its width, was hence five times greater in L3(GlcUAOA) than in L3(CTR) or L3(EtOH) (Figure 2C). *C. officinalis* glucuronides appeared to affect the integrity of the nematode larvae, reflected in changes in larval morphology.

### 2.2. HPLC Profile of H. polygyrus bakeri L3 Larvae Exposed to C. officinals Saponins

The water-soluble somatic protein extracts of *H. polygyrus bakeri* L3 were measured using high performance liquid chromatography (HPLC). Different chromatograms were observed for L3(CTR), L3(EtOH) and L3(GlcUAOA) (Figure 3), with each group of larvae displaying a unique elution profile representing a unique molecular pattern.

The L3(GlcUAOA) chromatogram displayed more peaks for the larger peptides and appeared to contain more peptides of different sizes than the control larvae, i.e., L3(CTR) and L3(EtOH). The highest peak shifted from 12 min in L3(CTR) to 15–18 min in L3(GlcUAOA), and two additional peaks were eluted at 10 min and at 33 min in L3(GlcUAOA). Figure 3 shows chromatograms at OD 254 nm. *C. officinalis* glucuronides appear to affect the HPLC profile of water-soluble somatic protein extracts.

### 2.3. Marigold GlcUAOA Affects Antigenicity of L3 Proteins

Figure 4 and Appendix A, shows SDS-PAGE protein pattern of. *H. polygyrus bakeri* L3 (A) and Western blot protein pattern of L3 recognised by hyperimmune serum of rabbit, immunised with L3 (CTR) (B). SDS-PAGE analysis found both ethanol and GlcUAOA treatment to alter the intensity of somatic L3 protein expression: bands in the range of 225–95 kDa, 77–73 kDa, 54–50 kDa, 42–40 kDa, 36–32 kDa, 27–18 kDa were weaker when compared to those obtained from control larvae. However, the bands of 16–10 kDa were of equal intensity to those of L3 (CTR) (Figure 4A).

In addition to this altered expression, the antigenicity of the L3 treated with EtOH and GlcUAOA was also found to be altered; after detection with hyperimmune rabbit serum, some of the bands observed in L3(CTR) were absent from the L3(EtOH) and L3(GlcUAOA) groups.

The immunoblots found that all groups of larvae displayed different reactive protein profiles, and inter-larva variations were detected. One of the most striking differences observed was the reduction in the number of positive bands: nine were displayed in L3(GlcUAOA) and 15 in L3(EtOH), while 19 were found in control larvae. 

A number of bands present in L3(EtOH) and L3(CTR), viz. 128, 50.8, 49.1, 42.7, 40.7, 38.1, 37.5, 24, 23.7 kDa, were not recognised in L3(GlcOAUA). In addition, the 122, 50.8, 49.1, 40.7, 38.1 and 23.7 kDa bands present in L3(EtOH) were not identified in L3(GlcOAUA). Degradation in proteins of parasitic antigens was observed following treatment with *C. officinalis* oleanolic acid glucuronides.

### 2.4. LC-MS/MS Identification of Antigenic Proteins of L3 H. polygyrus bakeri

The L3(CTR) somatic extract was run on 12% SDS-PAGE to separate the constituent proteins and stained with Coomassie blue G250. Four protein bands that corresponded to the “missing” bands on the L3(GlcUAOA) immunoblot recognised by hyperimmune rabbit serum were identified: 50.8 kDa (1), 49.1 kDa (2), 42.7 kDa (3) and 40.7 kDa (4) (Figure 4 and Appendix A). These bands were carefully excised from the L3(CTR) control gel and further analysed using gel tryptic digestion in conjunction with LC/MS/MS. Proteomic analysis of *H. polygyrus bakeri* L3 somatic extract was performed according to an extensive proteomic dataset, and a total of 25 proteins were identified in these four bands by LC–MS/MS.

As a result, 18 potentially antigenic proteins were identified. Five proteins were present in more than one band. The greatest number, ten proteins, were present in bands 1 and 2: these were analysed together due to their close proximity to one another. Five of the proteins present in bands 1 and 2 were also identified in bands 3 and 4: ATP synthase F1, beta subunit; paramyosin isoform 1; actin variant 1 and heat shock 70 protein. The hypothetical protein Y032 was noted in different sequence variants; these were classified by Uniprot ID analysis as V-type ATPase, beta subunit; protein unc-87; arginine kinase and fructose- bisphosphate aldolase class. The proteins were present in separate bands as polypeptide fragments with different relative molecular weights. As the proteome of *H. polygyrus bakeri* is insufficiently understood, most of the peptides were assigned to the proteins of related species of nematodes, having a broad spectrum of known protein sequences (Table 1, Appendix A).

### 2.5. Functional Categories of Potentially Antigenic Proteins of L3 H. polygyrus bakeri by Gene Ontology Analysis

The proteins of L3(CTR) that were less abundant in the gel for L3(GlcOAUA) were categorised according to their molecular function, cellular component and biological processes by gene ontology (GO). The GO annotations for candidate *H. polygyrus bakeri* L3 complex proteins classified 17 proteins for biological processes, 14 for molecular function and 21 for cellular components. More detailed information is shown in Figure 5.

The functions of identified proteins are described and presented in (Appendix A).

*C. officinalis* GlcUAOA treatment, known to induce apoptotic processes within parasite cells, strongly affected the subcellular morphology and protein expression of *H. polygyrus bakeri* L3, indicated by the fact that hyperimmune rabbit serum recognised fewer nematode specific proteins.

## 3. Discussion

*C. officinalis* oleanolic acid glucuronides (GlcUAOA) have been shown to affect the development of free-living stages of *H. polygyrus bakeri* and reduce their infectivity [20]. GlcUAOA was found to impair the ultrastructure of *H. polygyrus bakeri* L3 larvae to a greater extent than ethanol, irrespective of ethanol exposure. GlcUAOA saponins may act both intracellularly or intercellularly and alter membrane permeability [27]. Changes were observed in the thickness and elevation or peeled off the cuticle layers, indicating that both ethanol and GlcUAOA deregulated cuticle synthesis. In nematodes, the cuticle is an extracellular matrix consisting predominantly of small collagen-like proteins that are extensively crosslinked [28]. In *Caenorhabditis elegans*, the cuticle did not present an absorption barrier for ethanol [29]. In the presence of ethanol collagen retains its triple helix [30]. Exposure of *H. polygyrus bakeri* larvae to ethanol resulted in swelling and separating of the layers. The saponins dissolved in ethanol seem to revert the ethanol effect. Saponins, due to the lipophilic/hydrophobic nature of their aglycon and interaction between the sugar chains, may induce pore formation, destroying cell membranes which may prevent separation of the layers. By relocation into the cytosol and affecting metabolic pathways, saponins may contribute to the leaky cuticle and destruction of larvae. We assume that glucuronides of oleanolic acid exhibited significant destructive effects on the larvaes’ ultrastructure and caused distinctive damage to body surfaces and internal structures. Saponins, by destroying the cuticle, or preventing its renewal, can reduce *H. polygyrus bakeri* infectivity in mice [20,26].

Condensed tannins are also known to inhibit or delay the exsheathment of trichostrogylid L3 larvae by binding to cuticle proteins [2]. In addition, saponins disrupt the lipid bilayer and affect the interaction between transmembrane proteins and the cytoskeleton [31]. These changes in the membrane cholesterol-content may result in impaired P-glycoprotein activity and impact the anthelmintic resistance [32].

Being producers of ATP via oxidative phosphorylation, mitochondria are an important target for many antiparasitic drugs. GlcUAOA treatment physically disrupted mitochondria, decomposed the cytosol and significantly changed the cell proteome; exposure to GlcUAOA resulted in unfolding of the inner membrane, with mitochondrial cristae being absent. Neither mitochondrial ATP synthase, carboxyl transferase or fructose-1, 6-bisphosphate aldolase, all of which facilitate energy production and oxidative metabolism, could be indicated on the L3 (GlcUAOA) immunoblot. Glutamate dehydrogenase was also negative; it is known to contribute to energy production in the Krebs cycle, and to enable redox homeostasis and cell signalling [33]. Changes in mitochondrial function associated with the ATP-dependent electron transport chain, glycolysis, or fatty acid oxidation have been recorded after exposure to toxins and saponins [34,35]. GlcUAOA treatment may alter protein metabolic activity, resulting in swollen mitochondria, an indicator of oxidative stress [36,37]. Mitochondrial dysfunction can increase reactive oxygen species (ROS) production and shorten lifespan: larvae exposed to GlcUAOA in agar culture demonstrated slower movement and died within two weeks (our observations). These observations should be taken into account when the safe saponin dose treatment of animals or humans is considered. Overdosing natural products, particularly in connection with long lasting applications, may influence cell condition and mammalian health. Among the 18 proposed proteins demonstrating weakened antigen-antibody binding affinity, seven possess an ATP-binding function (mitochondrial ATP synthase subunit beta; myosin head; vacuolar H+-ATPase (V-ATPase), beta subunit; heat shock protein 70; actin, putative chaperone protein DnaK; arginine kinase). V-ATPase is an ATP-dependent proton pump expressed in the hypodermis, intestine and H-shaped excretory cells of *Caenorhabditis elegans*. V-ATPase plays an essential role in nematode nutrition, osmoregulation, cuticle synthesis, neurobiology and reproduction [37] and is known to regulate the pH of the intracellular compartments and extracellular space. Any changes in its molecular specificity caused by exposure to saponin would also affect other cellular processes, such as intracellular membrane traffic, protein processing and degradation, and the coupled transport of small molecules and ions [35]. As V-ATPase appears to be a desirable target for anti-nematode drugs [38], its deactivation by GlcUAOA indicates specific antiparasitic potential. Interestingly, saponins have previously been found in all cell structures [39] and our results suggest that GlcUAOA interferes with the proteins influencing the oxidative metabolism of larvae.

These structural changes impair the integrity of the plasma membrane, weaken its molecular characteristics and reduce cell viability [40]. Several cytoskeletal proteins critical for larval locomotion may be affected by GlcUAOA treatment, including CRE-UNC-15, myosin, actin, tropomyosin and paramyosin. CRE-UNC-15, i.e., nucleoside-triphosphate phosphatase, also positively regulates sarcomere organisation. It is found in the skeletal muscle myosin thick filament assembly, where it interacts selectively and non-covalently with all cytoskeleton protein components, including actin, microtubule, or intermediate cytoskeleton filaments (https://www.ebi.ac.uk/QuickGO/term/GO:0008092) (accessed on 28 February 2019).

Immunoblotting and MASCOT analysis revealed that saponin exposure altered the immunogenic characteristics of both tropomyosin and actin proteins. Tropomyosin influences locomotion and regulates protein binding [41]. Myosin itself is typically arranged in thick filaments, participates in muscle contraction and appears to be a critical protein for pharyngeal pumping and locomotion. The myosin head, specific to *Haemonchus contortus*, is involved in the response to stress [42]. Tropomyosin also acts as a target for the immune response as it is expressed on the surface of parasitic nematodes [43,44]. It is possible that dysfunctions in the cellular cytoskeleton were responsible for the reductions in motor activity of *H. polygyrus bakeri* larvae observed following GlcUAOA treatment [16]. GlcUAOA exposure led to dysfunctions in the cuticle layers; this may be associated with the action of tropomyosin, which plays a role in the catabolism of macromolecules in the cell wall during moulting. GlcUAOA also prevented protein recognition by hyperimmune rabbit serum and therefore the fact that GlcUAOA appears to reduce the potential of tropomyosin to suppress the host immune system [45] may offer promise in preventing parasite invasion. In addition, protective immunity may also be modulated by the galectin domain containing protein (GDC) present in larval stages [46].

Protein profiles change substantially during the nematode life cycle, and these are crucial points in transition from the free living to the parasite stage [47]. In the parasite-host interaction, several enzymes showing antigenic properties are also associated with a range of non-glycolysis functions: fructose-1, 6-bisphosphate aldolase participates in adhesion to host cells, plasminogen binding and successful invasion [48]. In addition, moonlighting proteins such as enolase are prerequisites for motility, adhesion and invasion. The loss of enolase expression delays the larval development of *Ascaris suum* [49]. Plasminogen supports parasite establishment in the host by binding to enolase in larvae [50]. Glutamate dehydrogenase possesses antigenic properties and has been linked to ammonia metabolism, acid-base equilibrium, redox homeostasis or lipid biosynthesis and lactate production [51,52].

Microtubules are dynamic filaments formed by the polymerisation of the heterodimeric protein α-/β-tubulin. They play an essential role in ensuring the correct spatial organisation of the cytoplasm, cell shape and polarity, cell division, intracellular transport and cell wall deposition [53]. Two of the proteins indicated by MASCOT, viz. beta-tubulin isotype 1 and the Tubulin/FtsZ family, both of the GTPase domain, demonstrate GTPase activity: they are structural constituents of the cytoskeleton and facilitate GTP binding [54].

A number of studies have examined natural compounds which exert cytotoxic properties by interacting with tubulin [55]; in addition, the microtubule complex has proven to be a suitable target for the development of anti-nematode therapeutic agents [56,57,58]. Therefore, our data would suggest that the combination of GlcUAOA with benzimidazoles, which inhibit microtubule polymerisation by binding to β-tubulin [59], may demonstrate a synergistic effect; this would allow resistance to be provided against parasites at lower doses, reducing the toxic effect of the drug on the host.

Immunoblotting and MASCOT analysis of the larval proteins found the beta-tubulin isotype to lose its epitope specificity upon exposure to GlcUAOA. Thus, like other triterpenoid saponins, GlcUAOA could interfere with the stability and polarisation of microtubules [60], resulting in mitotic block and apoptosis [61,62,63,64]. This suggests that the GlcUAOA treatment induced microfiber/microfilament fragmentation, reflected in the HPLC profile of the nematode proteins. Additionally, new peaks appeared early in elution following saponin treatment: these may have been associated with small nuclear or cytosolic proteins that may have leaked following larval membrane permeabilisation, pore formation and apoptosis [20,63,65], as well as soluble intracellular materials that may have diffused away from the cell. Greater numbers of peaks were found in the HPLC profile of the somatic proteins of the affected larvae, which was characteristic of protein fragmentation and dissociation. In addition, in larvae exposed to GlcUAOA extract, the cytoskeletal structures were found to be shortened and aggregated, which might result in their different position at SDS-PAGE gel. The shortening of the cellular myofibrils observed following treatment with *C. officinalis* GlcUAOA is an example of a stress-related change.

Changes in the cuticle, hypodermis, muscle cells, nuclei and mitochondria are all indicative of apoptosis. Similar changes are observed in tumor cells treated with triterpenoid saponins [66] and aqueous *C. officinalis* extract [67]. Further, tannin-rich plant extracts have been found to induce the formation of numerous vesicles and degenerative changes in the hypodermis and cytoplasm of infective larvae of ruminants, these being typical of the death of muscular and intestinal cells [68]. Saponin application results in the permeabilisation of cells through the disruption of actin filaments, and this can accelerate the loss of cellular macromolecules. MASCOT analysis confirmed that the protein bands containing actin in the L3 electrophorograms lost their antigen-antibody binding affinity after exposure to GlcUAOA, suggesting that these proteins may have undergone degradation.

Arginine kinase has been observed within the excretory-secretory protein profile of *H. polygyrus bakeri* [69] and is known to be involved in host immune responses and to exhibit distinct immunomodulatory properties [70]. Arginine kinase specific to *Ancylostoma ceylanicum* catalyses the ATP-phosphorylation of L-arginine [71]. Its susceptibility to GlcUAOA suggests that other molecules crucial for invasion of larvae into the host intestine may also be selectively deactivated. Such disruption of enzyme activity by GlcUAOA may result in a weakened L3 response to stressful conditions and reduced infectivity [72].

Exposure to *C. officinalis* GlcUAOA resulted in 18 proteins in the *H. polygyrus bakeri* L3 controls losing recognition by hyperimmune rabbit serum. Of these, the predominant families were structural proteins, e.g., actin, tropomyosin and beta-tubulin, with highest sequence similarity to corresponding proteins of *Necator americanus* and other nematode species. GlcUAOA exposure was associated with dysfunctions in the myosin, myosin head and myosin tail protein families taking part in signalling and cellular processes, such as cytoskeleton proteins, actin filaments or microfilaments and actin-binding proteins. In addition, members of the ATP synthase F1 subunit related to *Necator americanus* were also associated with such dysfunctions. *C. officinalis* GlcUAOA treatment also changed the glycosylation pattern of *H. polygyrus bakeri* larvae, and this had a profound effect on the immune response in infected mice: proinflammatory cytokine level increased, but the number of immunogenic proteins recognised by hyperimmunne rabbit serum decreased [20]. Changes in the immunogenic protein profile of larvae exposed to GlcUAOA have implications on the success of nematode invasion in mice [26,72]. Ensheathed L3 *H. polygyrus bakeri* have been found to induce specific IgG production [73]; in addition, the somatic antigens of ethanol-treated L3 were recognised differently by anti *H. polygyrus bakeri* L3 hyperimmunne rabbit serum compared to the saponin-treated L3: several proteins expressed by L3 (GlcUAOA) demonstrated lower antigen-antibody binding affinity than controls. These findings suggest that saponin exposure did affect certain protein-specific characteristics and consequently larval immunogenicity.

Saponins can be used to facilitate the passage of drug molecules or other natural products through the cell membrane by increasing its permeability. This ability makes saponins promising natural products in pharmacological and medical research and therapy, particularly as agents for enhancing drug efficacy. *C. officinalis* GlcUAOA may be categorised as exerting a negative regulatory influence on nematode development, as it significantly affects the development and ultrastructure of *H. polygyrus bakeri* larvae and dysregulates the immunogenicity of several proteins.

Our findings pave the way for more precise determination of the individual proteins affected by saponin exposure using 2D electrophoresis.

## 4. Materials and Methods

### 4.1. Separation of Oleanolic Acid Glucuronides

Air-dried *C. officinalis* flowers were extracted with boiling methanol (three times for 30 min). After adding an equal volume of water to the combined extracts, the methanol was removed by distillation in a vacuum rotary evaporator, and the aqueous residue was extracted with *n*-butanol; the mixture was evaporated to dryness, re-dissolved in methanol and precipitated with cold ethyl acetate. The precipitate was centrifugated at 6000 rpm for 10 min. The pellet was resuspended with the use of Potter homogenizer and centrifuged again. Glucuronides of oleanolic acid GlcUAOA were purified on a silica gel column eluted with a mixture of chloroform/methanol (95:5, 90:10) and chloroform/methanol/water (30:10:1). The fractions containing the oleanolic acid glucuronides were combined. The purity of GlcUAOA (90 %) was verified by TLC and HPLC [26]. The powder was dissolved in 96% ethanol before use.

### 4.2. Exposure of H. polygyrus bakeri to Oleanolic Acid Glucuronides

*H. polygyrus bakeri* L3 was cultured as previously described [20]. Briefly, the nematode eggs were isolated from faecal material of infected C57BL/6 mice by the flotation technique. The eggs were extensively washed with water and PBS (pH 7.2), then split into glass plates containing 5 mL of Nematode Growth Medium (NGM) agar with *Escherichia coli* strain OP50 added as a food source [74], as well as into NGM containing 100 μg × mL^–1^ GLcUAOA. As the GLcUAOA was dissolved in ethanol, an additional control was prepared consisting of plates of NGM containing an equivalent concentration of 8% ethanol (EtOH). The negative control larvae (CTR) were cultured on agar. Third-stage larvae (L3) were harvested after 10 days. The viability of the parasites exposed to ethanol and GLcUAOA was evaluated microscopically on day 10 of agar culture. Three groups of infective larvae were examined: L3 (CTR), L3 (EtOH) and L3 (GLcUAOA) and were used for TEM and HPLC analysis. 

### 4.3. Evaluation of L3 by Transmission Electron Microscopy

Nematodes collected from agar culture were washed five times in fresh distilled water and used for ultrastructural observation by transmission electron microscopies. The nematodes were fixed with 2.5% glutaraldehyde cacodylic buffer, incubated for one hour, then washed in 0.1M cacodylic buffer. Next, the samples were postfixed in 1% OsO_4_ in ddH_2_O for one hour and washed three times in ddH_2_O. After postfixation, the samples were dehydrated through a graded series of EtOH (50%—10 min, 70%—24 h, 90%—10 min, 96%—10 min, anhydrous EtOH—10 min, acetone—10 min) and infiltrated with epon resin in acetone (1:3—30 min, 1:1—30 min, 3:1—2 h), infused twice for 24 h in pure epon resin and polymerised at 60 °C for 48 h [75]. Next, 60 nm sections were prepared with an ultramicrotome RMC MT-X (Boeckeler Instruments, Inc., Tuscon, AZ, USA), contrasted with uranyl acetate and lead citrate and examined on a Zeiss LEO 912AB electron microscope (ZEISS, Oberkochen, Germany). Images were captured by a Proscan high-speed slow-scan CCD camera (using EsiVision Pro 3.2 software (Soft Imaging Systems GmbH, Münster, Germany). The procedure was performed according to Reynolds (1963) [76].

### 4.4. Protein Sample Preparation and HPLC Profile

The somatic protein profiles of the L3 larvae were analysed with an Alliance 2695 High Performance Liquid Chromatography (HPLC) unit coupled to a Waters photodiode array detector, and a ProteinPak (Waters, Milford, MA, USA) column was used to separate proteins according to molecular size [77]. At least 50,000 larvae were lysed in 0.5 mL of PBS on ice using ultrasound. A protease inhibitor cocktail (Sigma-Aldrich, Darmstadt, Germany) was added. The samples were then centrifuged at 18,000 g for five minutes at 4 °C and then stored in −80 °C until use. Protein concentration was measured using the Bradford protein assay (Bio-Rad Laboratories, Inc., Hercules, CA, USA). Protein samples of 100 μL concentrated to 1 mg/mL were loaded on a column and eluted isocratically with PBS (pH 7.4) with a flow rate of 400 μL/minute for 45 min. Spectra were collected in the range 190–650 nm. HPLC fractioning experiments were calibrated with synthetic peptides to allow comparisons between experiments. Data was analysed with the Empower software (Waters, Milford, MA, USA). Representative chromatograms were taken at 254 nm at chosen time points.

### 4.5. Preparation of L3 Protein Samples

*H. polygyrus bakeri* larvae: L3(CTR), L3(EtOH) and L3(GlcOAU) were washed extensively and 2000 nematodes were resuspended in 200 μL of PBS (pH 7.2, Gibco), sonicated on ice using Vibra Cell^TM^ (Sonics and Materials Inc., Newtown, CT, USA) and centrifuged 10 min at 10,000 g. Soluble proteins were separated by sodium dodecyl sulphate-polyacrylamide gel electrophoresis (SDS-PAGE) in a Bolt^®^ Mini Gel Tank system (Novex, Life Technologies, Carlsbad, CA, USA). Larval proteins were mixed with reducing sample buffer, heated for 10 min at 70°C in a water bath and then loaded as four replicates (20 μL containing 10 μg of protein per well) on Bolt 4–12% Bis-Tris Plus 15 well precast gel. Proteins were visualised by Coomassie Brilliant Blue G-250 (Sigma-Aldrich, Darmstadt, Germany) staining. Electrophoresis was performed according to manufacturer’s instructions; 35 min, 165 V in 1X MES Running Buffer. 

The protein bands were transferred onto a nitrocellulose membrane in an iBlot^®^ Dry Blotting System (Novex, Life Technologies, Carlsbad, CA, USA) for six minutes at 25V. Following this, the membrane was blocked with 1% BSA (bovine serum albumin), (Sigma-Aldrich, Darmstadt, Germany) for two hours at room temperature.

### 4.6. Recognition of H. polygyrus bakeri L3 Immunogenic Proteins

Immune sera to *H. polygyrus bakeri* L3 were prepared by immunising a New Zealand rabbit with L3 somatic extract in Freund’s complete adjuvant. The nematode larvae L3 were harvested from copro-culture on the 10^th^ day. Larvae were intensively washed in distillated water and in PBS (pH 7.2), and sonicated on ice. The total time of sonication was 10 min; 40% output for 10 s pulse and 30 s rest, fifteen times. The protein concentration was measured, extraction yield: 1000 L3 were equivalent to 300 µg of proteins. One 12-month-old rabbit was injected subcutaneously with 0.15 mg proteins of L3 somatic extract, at weekly intervals for four weeks. The blood was collected from the ear vein 21 days after the final inoculation. The blood sample was clotted then the separated serum was frozen and stored at −80 °C until use.

Protein samples isolated from *H. polygyrus bakeri* L3(CTR), L3(EtOH) and L3(GlcUAOA) were separated by SDS-PAGE and then analyzed by Western blot with rabbit immune anti-nematode L3 sera, diluted 1:1000. The membrane was incubated with goat anti-rabbit IgG1 conjugated with horseradish peroxidase (Santa Cruz Biotechnology, Inc., Dallas, TX, USA) diluted 1:4000. The bound antibodies were visualised by incubation with DAB (2,4-diaminobutyric acid, Sigma–Aldrich) and 0.03% (w/v) hydrogen peroxide. The L3 protein patterns were compared, and any immunoblot bands which were not recognised by rabbit immune sera in L3(GlcUAOA), i.e., were not visible from the immunogram, were localised on the L3(CTR) SDS-PAGE gel.

### 4.7. LC-MS/MS Identification

Selected protein bands with relative masses: 50.8–49.1(1 and 2), 42.7(3) and 40.7 (4) were cut from the SDS-PAGE gel of the L3(CTR) path. Bands of interest were excised from the 1D gels using sterile disposable scalpel blades and then subjected to trypsin digestion. Gel pieces were washed three times in 100 µL of 50 mM ammonium bicarbonate, 50% (*v/v*) methanol and then twice in 100 µL of 75% (*v/v*) acetonitrile, before drying. The gel pieces were rehydrated with a trypsin solution (20 mg trypsin/mL 20 mM ammonium bicarbonate), and incubated for four hours at 37 °C. Peptides were extracted from the gel pieces by washing twice in 100 μL of 50% (*v/v*) acetonitrile/0.1% (*v/v*) trifluoroacetic acid, before being transferred in solution to a fresh 96-well plate and dried before mass spectrometry analysis. All peptide samples were separated on an LC system (Famos/Switchos/Ultimate, LC Packings, Amsterdam, Netherlands) using water containing 0.1% TFA as the mobile phase. Following this, they were transferred to a nano-HPLC RP-18 column (nanoACQUITY UPLC BEHC18; Waters Associates, Milford, MA, USA) using an acetonitrile gradient (0–60% ACN) in the presence of 0.05% formic acid with a flow rate of 150 μL/min and analysed by electrospray ionization (ESI) Orbitrap mass spectrometry. A blank run preceded each analysis.

### 4.8. Data Analysis

The LC-MS/MS data were analysed against the NCBI database using Mascot Daemon (2.4.1, Matrix Science, London, UK), (http://www.matrixscience.com) (accessed on 7 January 2015), allowing a maximum of one miscleavage. The database was searched against Nematoda (379,686 sequences). Carbamidomethyl (C) was set as the fixed modification, while Carboxyethyl (K) and Oxidation (M) were included as variable modifications. Peptide mass tolerance and fragment mass tolerance were set as ± 30 ppm and ± 0.1 Da, respectively. Ion score cut-off was set above 50. The results were filtered using Mascot Percolator (http://www.matrixscience.com/) (accessed on 18 February 2019).

Gene Ontology (GO) annotation was performed using BLAST2GO (version 5.2, https://www.blast2go.com/blast2go-pro/download-b2g) (accessed on 18 February 2019), to specify the molecular functions, biological processes and cellular components for each protein. The NCBI BLASTp database was searched using default parameters (NCBI https://www.ncbi.nlm.nih.gov/; (BLASTp https://blast.ncbi.nlm.nih.gov) (accessed on 18 February 2019). The identified proteins were categorised by their molecular function, cellular component and biological processes according to gene ontology information obtained from UniProtKB (http://web.expasy.org/docs/swiss-prot_guideline.html) (accessed on 18 February 2019) and QuickGO (http://www.ebi.ac.uk/QuickGO/) (accessed on 28 February 2019) databases.

The study presents the results obtained from a representative experiment of three independent studies, as well as their statistical evaluation. The significance of differences between groups was determined by analysis of variance (ANOVA) using MINITAB Software (Minitab Inc., State Collage, PA, USA). All values are expressed as mean ± SE. A *p*-value <0.05 was considered to be statistically significant.

## Figures and Tables

**Figure 1 pathogens-10-00296-f001:**
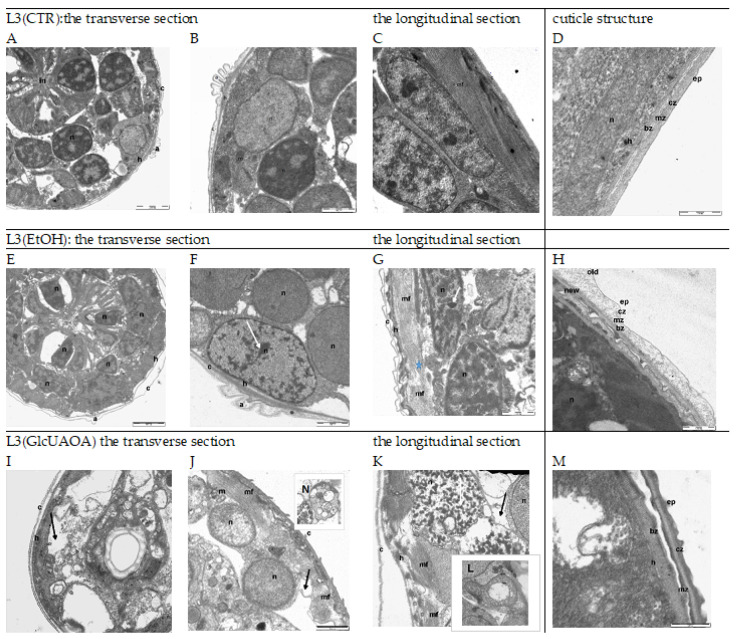
The ultrastructure of *H. polygyrus bakeri* L3 following marigold GlcUAOA treatment. Representative TEM images from morphological observations: Images (**A**–**D**)—L3(CTR) as negative controls with no apoptotic changes; Images (**E**–**H**)—L3(EtOH) exposed to ethanol; Images (**I**–**N**)—L3(GlcUAOA) demonstrating cell, apoptosis, nuclear condensation (white arrows) and cell shrinkage due to apoptosis (black arrows); autophagosome (**N**); the blue asterix at (G) indicates fragmented myofibrils; Image **L**—swollen mitochondrion of L3(GlcUAOA). a: alea; bz: basal zone; c: cuticle; cz: cortical zone; ep: epicuticle; h: hypodermis; im: inner membrane; in: intestine; m: mitochondrion; mf: myofibrils; n: nucleus. Scale bar: 2000 nm (**A**,**E**,**I**); 1000 nm (**B**,**C**,**E**–**G**,**J**–**I**) and 500 nm (**D**,**H**, **L**–**N**).

**Figure 2 pathogens-10-00296-f002:**
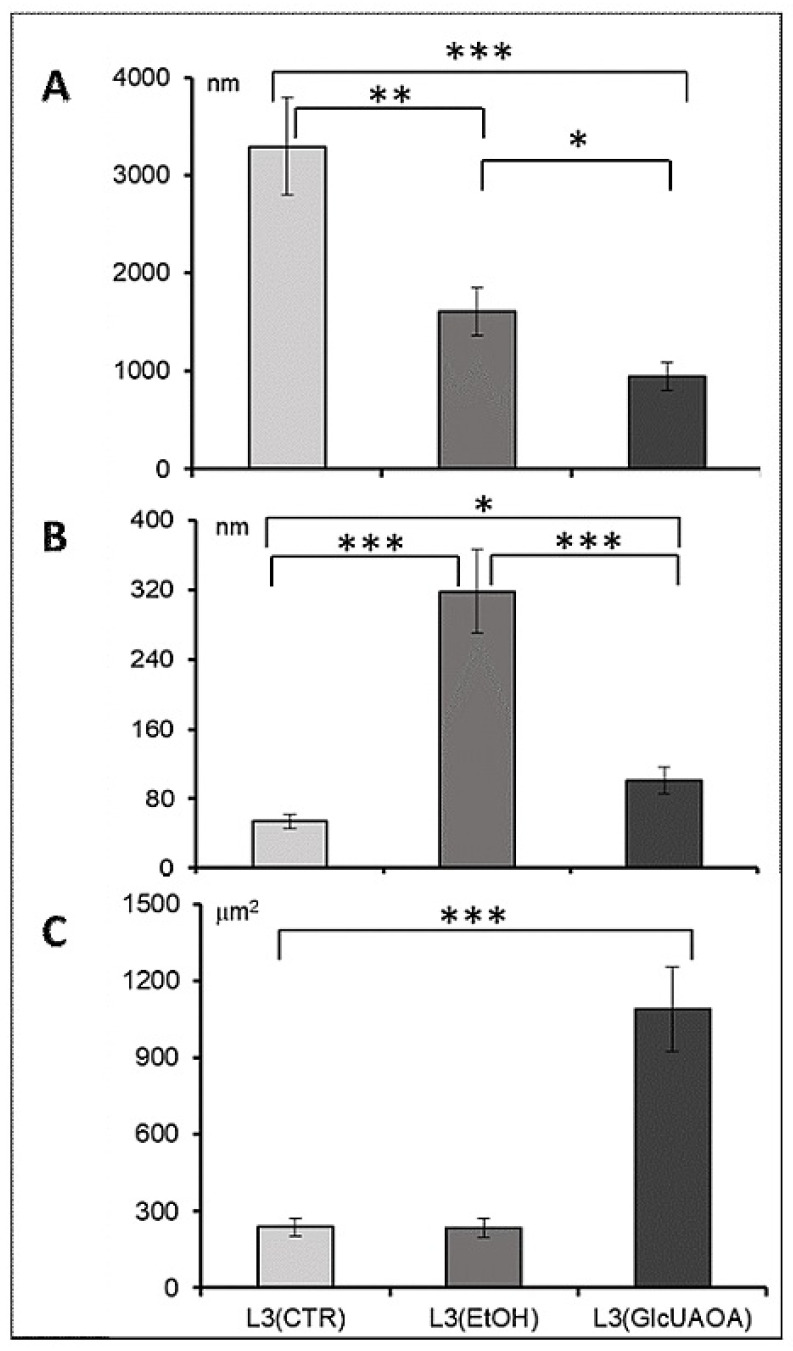
Morphometric analysis of cell structure in transverse and longitudinal section of *H. polygyrus bakeri* larvae; the length of myofibrils (**A**) in longitudinal section, the distance between the cuticle layers (**B**) in transverse section, the size of mitochondrion (**C**). Statistics: ANOVA, n = 10, * *p* < 0.05, ** *p* < 0.01, *** *p* < 0.001, mean ± SD.

**Figure 3 pathogens-10-00296-f003:**
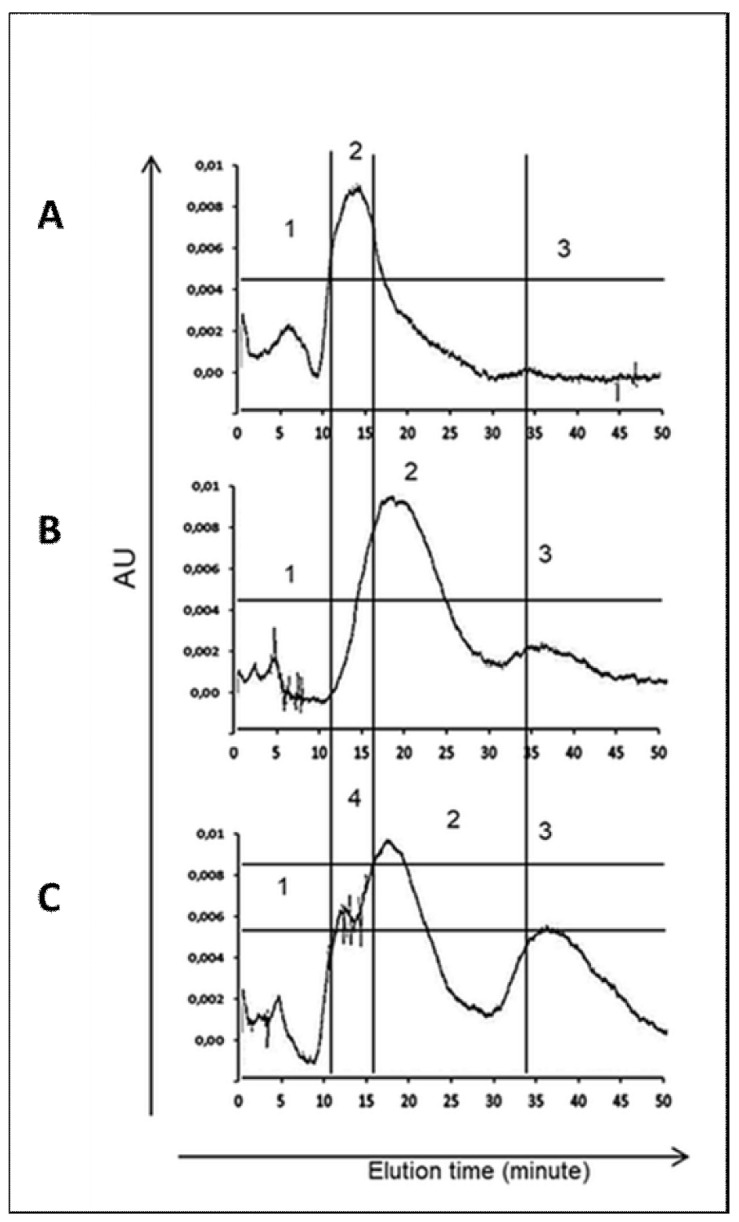
Elution profile of somatic protein isolated from *H. polygyrus bakeri* larvae. (**A**)—negative control L3(CTR); (**B**)—exposed to 8% ethanol L3(EtOH); (**C**)—exposed to *C. officinalis* glucuronides L3(GlcUAOA). A total of 100 µL of antigen solution was separated on a ProteinPak column and eluted isocratically using PBS (pH 7.4) with flow rate 400 µL/min for 45 min.

**Figure 4 pathogens-10-00296-f004:**
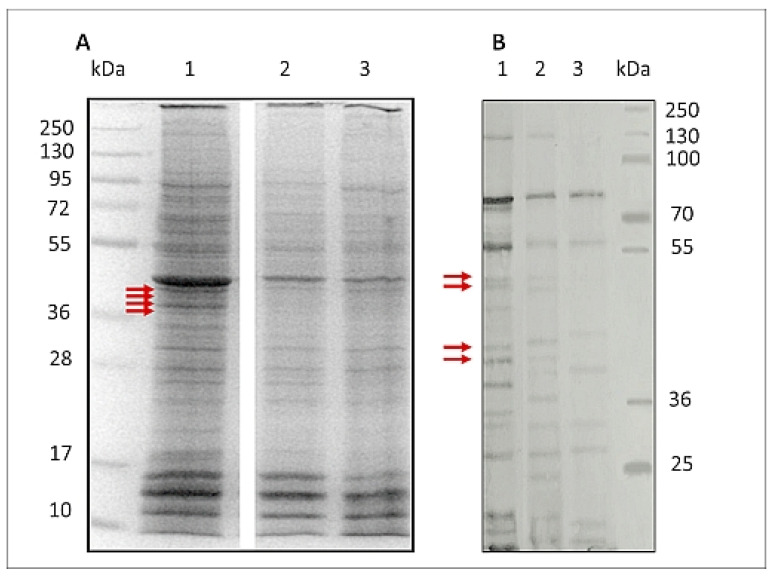
SDS-PAGE protein pattern stained with Coomassie blue of *H. polygyrus bakeri* L3 (**A**) and Western blot protein pattern of L3 (**B**) recognised by hyperimmune rabbit serum. Proteins extracted from: **1**—control larvae L3(CTR), **2**—larvae exposed to 8% ethanol L3(EtOH), **3**—larvae exposed to *C. officinalis* glucuronides L3(GlcUAOA): kDa—molecular mass of protein marker. Primary Western blot detection for B is placed in Appendix A. Red arrows indicate bands analysed by LC-MS/MS.

**Figure 5 pathogens-10-00296-f005:**
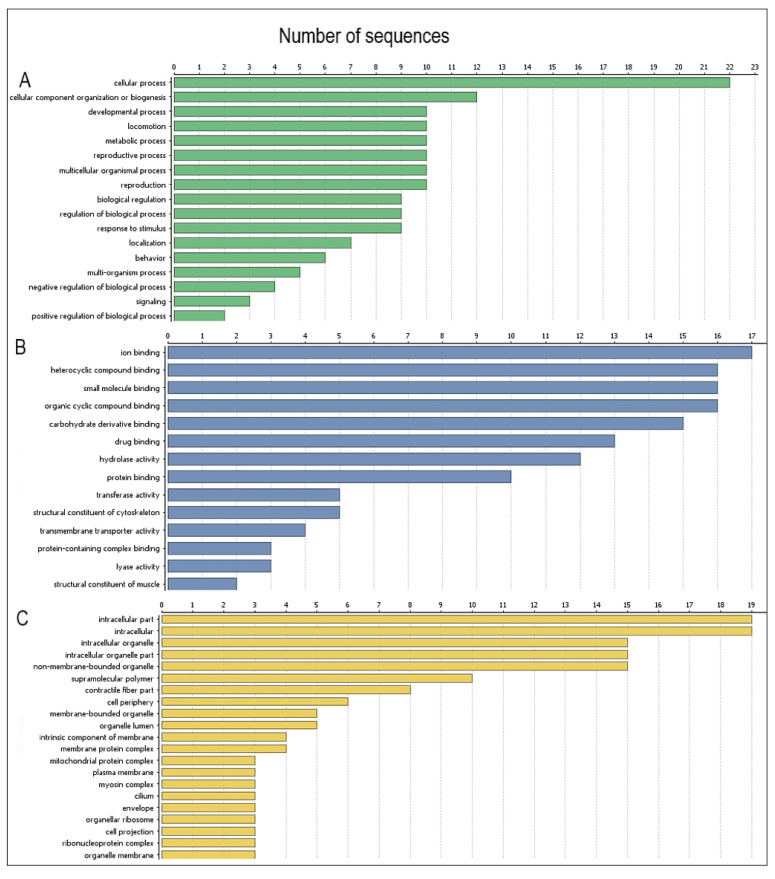
Functions of *H. polygyrus bakeri* L3 stage proteins according to Gene Ontology (GO): (**A**)—proteins categorised by their biological processes; (**B**)—proteins categorised by their molecular function; (**C**)—proteins categorised by their component category. Information obtained from UniProtKB and Quick GO databases.

**Table 1 pathogens-10-00296-t001:** *H. polygyrus bakeri* L3 proteins identified by hyperimmune rabbit serum, which were deactivated by *C.officinalis* GlcOAUA.

Band (kDa)	Accession (Gi Number)	Protein Name	Species	Score	Mass (kDa)	emPAI	Coverage %	Number of Significant Peptide Matches	Uniprot ID	Description	Length (aa)
1 and 2 (50.8-49.1)	gi|568290660	ATP synthase F1, beta subunit	*Necator americanus*	1325	58,151	1.57	0.45	28	XP_013300832.1	ATP synthase subunit beta, mitochondrial	541
	gi|160415837	Beta-tubulin isotype 1	*Cylicocyclus nassatus*	671	50,444	0.95	0.27	14	ABX39216.1	Beta-tubulin isotype 1	448
	gi|560120094	Myosin and Myosin head and Myosin tail domain containing protein	*Haemonchus contortus*	644	226,702	0.28	0.09	15	CDJ95285.1	Myosin head	1963
	gi|47606682	Paramyosin isoform 1	*Dictyocaulus viviparus*	620	100,878	0.65	0.22	13	AAT36324.1	CRE-UNC-15 protein	876
	gi|597882604	Hypothetical protein Y032_0003g1332	*Ancylostoma ceylanicum*	492	50,904	0.79	0.3	9	EYC31965.1	V-type ATPase, B subunit	458
	gi|371486394	Alpha tubulin	*Ostertagia ostertagi*	465	5053	0.79	0.24	11	AEX31242.1	Tubulin/FtsZ family, GTPase domain protein	446
	gi|568269548	Carboxyl transferase domain protein	*Necator americanus*	321	58,645	0.24	0.11	6	ETN70891	Carboxyl transferase domain protein	535
	gi|9438176	Heat shock 70 protein	*Parastrongyloides trichosuri*	318	70,452	0.35	0.17	6	AAF87583.1	Heat shock protein 70	644
	gi|162280611	Actin variant 1	*Dictyocaulus viviparus*	274	42,129	0.65	0.29	5	ABX82966.1	Actin	376
	gi|253721983	Glutamate dehydrogenase, partial	*Haemonchus contortus*	210	59,444	0.33	0.09	5	ACT34055.1	Glutamate dehydrogenase	532
3 (42.7)	gi|162280611	Actin variant 1	*Dictyocaulus viviparus*	2471	42,129	2.31	0.46	57	ABX82966.1	Actin	376
	gi|560127002	Lipoma HMGIC fusion partner protein and Heat shock protein 70 domain containing protein	*Haemonchus contortus*	771	9216	0.66	0.19	17	CDJ88360.1	Putative chaperone protein DnaK	829
	gi|301015486	Enolase	*Haemonchus contortus*	497	47,273	1.04	0.3	9	ADK47524.1	Phosphopyruvate hydratase	434
	gi|597877220	Hypothetical protein Y032_0010g849	*Ancylostoma ceylanicum*	374	62,832	0.6	0.11	9	EYC26588.1	Protein unc-87	558
	gi|47606682	Paramyosin isoform 1	*Dictyocaulus viviparus*	310	100,878	0.18	0.07	6	AAT36324.1	CRE-UNC-15 protein	876
	gi|157326537	Tropomyosin	*Heigmosomoides polygyrus*	291	3313	1.14	0.18	6	ABV44405.1	Tropomyosin	284
	gi|568290660	ATP synthase F1, beta subunit	*Necator americanus*	217	58,151	0.34	0.19	4	XP_013300832.1	ATP synthase subunit beta, mitochondrial	541
4 (40.7)	gi|325516326	Disorganized muscle protein 1	*Haemonchus contortus*	414	35,825	0.26	0.14	5	ADZ24723.1	Disorganized muscle protein 1	321
	gi|162280611	Actin variant 1	*Dictyocaulus viviparus*	411	42,129	0.82	0.37	10	ABX82966.1	Actin	376
	gi|560124746	Galectin domain containing protein	*Haemonchus contortus*	333	40,642	0.23	0.1	9	CDJ90629.1	Galactoside-binding lectin	356
	gi|541045947	Fructose-bisphosphate aldolase 2	*Ascaris suum*	316	45,469	0.1	0.06	6	ADY45824.1	Fructose-bisphosphate aldolase class-I	417
	gi|597874384	Hypothetical protein Y032_0015g2816	*Ancylostoma ceylanicum*	295	40,719	0.36	0.11	7	EYC23757.1	Arginine kinase	359
	gi|568290660	ATP synthase F1, beta subunit	*Necator americanus*	255	58,151	0.55	0.24	6	XP_013300832.1	ATP synthase subunit beta, mitochondrial	541
	gi|157381686	Heat shock protein 70	*Angiostrongylus vasorum*	253	5614	0.35	0.13	5	ABV46675.1	Putative chaperone protein DnaK	507
	gi|597834320	Hypothetical protein Y032_0324g2520	*Ancylostoma ceylanicum*	228	30,314	0.15	0.09	4	EYB84003.1	Fructose-bisphosphate aldolase class-I	282

## Data Availability

All data generated and analysed during these studies are included in this published article and in the Appendix A. The mass spectrometry proteomics data have been deposited to the ProteomeXchange Consortium via the PRIDE partner repository with the dataset identifier PXD024205 and 10.6019/PXD024205 [78].

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
