# Peer review of "Calendula officinalis Triterpenoid Saponins Impact the Immune Recognition of Proteins in Parasitic Nematodes"

_pathogens, 2021, doi:10.3390/pathogens10030296_

Round 1

Reviewer 1 Report

This study by Doligalska et al is focused on the effects of saponins from Calendula officinalis on parasitic nematodes. They specifically examined the effects of glucuronides of oleanolic acid on the ultrasctructure of H. polygyrus L3 larvae and their infectivity. Overall the study is interesting but there are some significant issues that need to be addressed.

Major Issues:

  1. The nematodes were exposed to 500µg of GLcUAOA by placing eggs onto 5mL plates and harvesting L3s after 10 days. So in this case all of the worms were constantly exposed to 500µg of GLcUAOA for 10 straight days. No mention of how many eggs were placed on the plates. Aside from the morphological changes that were measured, did this exposure affect the lifespan, infectivity, or reproduction of the nematodes? How does this kind of exposure, constant exposure to 500µg of GLcUAOA, translate into something potentially useful in the field?
  2. The authors draw strong conclusions from the qualitative appearance of bands on an SDS gel and a western blot in Figure 4. This leads to the important question of how much protein was loaded into each well of the gels. This is important, since the difference between treatments (L3(CTR), L3(EtOH), and L3(GlcUAOA)) may be in gene expression or protein translation. In the methods section 4.5, the authors only make reference to the volume of sample loaded, not the amount of protein. Figures 1 and 2 already establish that there are physical differences in the length of myofibrils and distance between cuticle layers. Could some of the protein differences pointed in out in Figure 4 be due to differences in size of the nematodes? How much total protein do the authors acquire from 2,000 nematodes of each treatment? Are they all L3s? Lastly, it is not clear in the legend the differences between panels A and B.
  3. The amount of protein is also important in the interpretation of the recognition of H. polygyrus L3 immunogenic proteins. Did the authors use the same starting amount of protein from each treatment? If not, the difference in protein amount may explain the difference in immogenic response by the host. Also, why did they use somatic extracts of the worms, why not excreted/secreted proteins (ESPs)? The ESPs are a major point of interaction between host and parasite. Whole worm homogenate seems less biologically relevant in an infection model (Hewitson et al. 2011; Heligmosomoides polygyrus Elicits a Dominant Nonprotective Antibody Response Directed against Restricted Glycan and Peptide Epitopes).
  4. Lines 205-207 make reference to an extensive transcriptomic data set that was used to identify proteins. Is this data set published or accessible? I’m uncertain about Pathogens policy on data availability, but it seems unhelpful to reference an unavailable dataset that was necessary to produce the results described in this paper.
  5. Lines 208-219, the authors refer to proteins “specific for…” some parasitic nematode. This language is confusing. What does this mean, aren’t all of the “missing” proteins from H. polygyrus? So then how are they specific to something like N. americanus? Do the authors mean that the protein sequence has highest similarity with other nematode species? This concern is linked to the “Species” column in Table 1. And again in the discussion when the authors expound on the various proteins, such as on lines 386 and 387.

Minor Issues:

  1. The life cycle is incorrectly described (lines 53-58), at least as I understand it. They mention that the nematodes hatch and go through two moults (L1 to L2, and L2 to L3) to get to L3 stage. They mention that the larvae then develop into L4 and L5 and then become adults. This would require 3 moults from L3 to adult (L3 to L4, L4 to L5, and L5 to adult). But L3s undergo 2 moults to become adults (L3 to L4 and L4 to adult). This should be corrected, or the authors can provide a citation that supports their claim of 5 larval stages for H. polygyrus. A citation should be used to support this life cycle, something like Roberts L.S., Janovy J., and Nadler S. 2013. Foundations of Parasitology, 9th Chapter 25:397-410. There’s a nice section there on Dictyocaulidae that describes the life cycle.
  2. Line 224 the authors make reference to bands that are absent from a gel. Since the gel is qualitative and no qPCR was done, they should probably refer to these proteins as less abundant. They may be on the gel but too faint to visualize.
  3. Line 233-258: what do we learn from this kind of analysis? It doesn’t seem reveal anything important to us.
  4. Lines 273-27: Are any of these effects concerns for mammalian health? Is GlcUAOA safe for mammals?

Minor comments and textual changes:

391: Spelling is incorrect for N. americanus.

Author Response

Dear Reviewer, thank you very much for very valid and constructive remarks. According to your suggestion, we improved the manuscript and clarified our results.

Reviewer 2 Report

The aim of the studies presented in the manuscript is to demonstrate effects of triterpenoid saponins in a nematode using transmission electron microscopy and biochemical studies including LC-MS/MS .

The correct name of the nematode is Heligmosomoides polygyrus bakeri. The Authors use in many places Heigmosomoides. For this I found 2, while for Heligmosomoides 11500 hits. Both forms are used in the text. The correct form should be used throughout.

Line 16 and many other places: the rabbit antibody used in the studies is referred to as IgG1, IgG, in the maniscript Ig fractionation is not mentioned. In other places rabbit serum is mentioned (line201). Correctly, hyperimmune rabbit serum was used, which most probably contained a mixture of immunoglobulins of different classes, including IgG1, IgG2a, and IgG2b. If the antibody was IgG1, the evidence for this should be presented and the fractionation should be described in the Materials and Methods section.

As the alcohol in itself and its combination with the saponin caused substantial changes as detected by TEM and also altered the protein expression profile, a toxicity curve would be essential.

Line 18: „disrupt the immunogenic specificity of certain L3 proteins”. This is incorrect phrasing. The specificity is not disrupted. Either the immunological epitope is altered or the protein is degraded or missing (downregulated?).

Line 93: An introductory sentence should be inserted on the treatment of the samples for the analysis.

Line 128: could autophagosomes be seen, or is there any direct evidence in the presented studies for autophagy?

Line 129: „The highest magnification at 500 nm…” – incorrect phrasing

Figure 2.: Ethanol caused a separation of the cuticle layers, but the combination of alcohol and the compound seems to revert this effect. What is the explanation for this phenomenon?

Section 2.2. HPLC profile. Figure 3. The shift shows again (similarly to Fig 2B) that alcohol treatment alters the protein composition of the water-soluble fraction and combination of alcohol with the compound will produce a third type of reaction/curve. Is it possible that there is a synergistic or inhibitory effect of the combination? Were the fractions analyzed with respect to composition?

Line 175: “rabbit antiserum immunized with…” mixed metaphor

Line 177: “a treatment to altered the intensity of…” ?

Line 185: “the immunogenicity of the L3 extracts treated with EtOH and GlcUAOA was also found to be altered; „ This is not the immunogenicity of the extracts. This is the reaction of the extracts with the hyperimmune serum. We speak about altered immunogenicity if the immune sera obtained after immunization with the different larval groups elicit a different expression pattern of proteins.

Line 191: “immune positive bands” ?

Line 195: "lower recognition". Incorrect phrasing: Lack of protein?, degradation?

Section 2.4.: These type of questions can be answered by the analysis of immunoprecipitated material or excised "spots" from 2D gels, as a band in SDS PAGE generally contains many proteins, including those, reacting with the antibody in W-blot analysis and comparative studies. With other words: the detection of a protein by MS does not necessarily mean that the same protein is visualized by W-blot, even though it is excised carefully.

Line 384: „L3 controls losing recognition by rabbit immune sera” - incorrect phrasing

Line 483: How was the immunogen prepared?

Author Response

Dear Reviewer, thank you very much for the revision of the paper. According to your suggestions, we improved the manuscript and clarify our results.

Reviewer 3 Report

This manuscript describes the effects of herb extracts (triterpenoid saponins from Calendula officinalis) on parasitic nematodes H. polygyrus bakeri larvae.  This nematode as a natural pathogen of mice provides a useful laboratory model to study immune responses to gastro-intestinal nematodes that infect humans and livestock. The novelty of the manuscript is to describe the ultrastructure changes of nematode L3 larvae after herb extracts treatment as well as changes in the profile of immunogenic proteins from their somatic extracts. Authors used up to date methodology involved TEM electron microscopy for description of morphological changes in nematodes and HPLC, immunoblotting and LC-MS/MS for separation and identification of proteins interacting with immune system of the host. Authors bring the evidence that triterpenoid saponins alter subcellular morphology of the nematodes. Using bioinformatics their describes also possible disruption of immunogenic specificity and bring the list of candidates.

Minor corrections:

  • Latin name of the nematode is Heigmosomoides or Heligmosomoides? The first one is used mostly in the text but the second also occurs (Lines 74, 536, 537).

Author Response

Dear Reviewer,

thank you for your opinion. According to your suggestion, we improved the manuscript. 

Round 2

Reviewer 1 Report

The revised version of the manuscript is significantly improved and I applaud the authors for the way they handled the reviewers' concerns and revised the paper. This will be of interest to other researchers in the field.

Author Response

Dear Reviewer,

We are grateful for all your suggestions and comments on our manuscript. In the attachment, you can find the corrected manuscript.

Maria Doligalska

Reviewer 2 Report

In the manuscript, lines 402, 405,506,  and in

Supplementary Fig.S. "Western blot protein pattern of L3 recognized by rabbit IgG1, immunized with ... ".

"IgG1" should be changed to "hyperimmunne rabbit serum", as the antibody was used as such.

The manuscript has been improved.

Author Response

Dear Reviewer,

We are grateful for all your comments on the manuscript. All our mistakes were corrected according to your suggestions. The manuscript is attached. 

Maria Doligalska